# Proof-of-Familiarity: A Privacy-Preserved Blockchain Scheme for Collaborative Medical Decision-Making

**Jinhong Yang [1,\*], Md Mehedi Hassan Onik [2]**  **, Nam-Yong Lee [3], Mohiuddin Ahmed [4] and Chul-Soo Kim [2]**

[1] Department of Healthcare and IT, Inje University, Gimhae 50834, Korea
[2] Department of Computer Engineering, Inje University, Gimhae 50834, Korea;
hassan@oasis.inje.ac.kr (M.M.H.O.); charles@inje.ac.kr (C.-S.K.)
[3] Department of Applied Mathematics, Inje University, Gimhae 50834, Korea; nylee@inje.ac.kr
[4] School of Science, Edith Cowan University, Perth 6027, Australia; m.ahmed.au@ieee.org
[\*] Correspondence: jinhong@inje.ac.kr; Tel.: +82-10-8518-4555

**Featured Application:** **The work facilitates a collaborative medical decision-making scheme among patients, doctors and insurance companies through a blockchain-based architecture.**

**Abstract:** The current healthcare sector is facing difficulty in satisfying the growing issues, expenses, and heavy regulation of quality treatment. Surely, electronic medical records (EMRs) and protected health information (PHI) are highly sensitive, personally identifiable information (PII). However, the sharing of EMRs, enhances overall treatment quality. A distributed ledger (blockchain) technology, embedded with privacy and security by architecture, provides a transparent application developing platform. Privacy, security, and lack of confidence among stakeholders are the main downsides of extensive medical collaboration. This study, therefore, utilizes the transparency, security, and efficiency of blockchain technology to establish a collaborative medical decision-making scheme. This study considers the experience, skill, and collaborative success rate of four key stakeholders (patient, cured patient, doctor, and insurance company) in the healthcare domain to propose a local reference-based consortium blockchain scheme, and an associated consensus gathering algorithm, proof-of-familiarity (PoF). Stakeholders create a transparent and tenable medical decision to increase the interoperability among collaborators through PoF. A prototype of PoF is tested with multichain 2.0, a blockchain implementing framework. Moreover, the privacy of identities, EMRs, and decisions are preserved by two-layer storage, encryption, and a timestamp storing mechanism. Finally, superiority over existing schemes is identified to improve personal data (PII) privacy and patient-centric outcomes research (PCOR).

**Keywords:** healthcare data security; clinical data sharing; blockchain; data privacy; shared decision-making; medical information sharing; PoF; cryptography; healthcare; distributed ledger

## 1. Introduction

Shared decision-making (SDM) has been used as an optimum way of health care decision sharing among healthcare practitioners. Collaborative decision-making happens when individuals (two or more) underwrite diverse expertise and skill to the decision-making mechanism [1]. Studies [2,3] have already inspired clinicians to use SDM while providing preventive health and medical verdicts. Collaborative medical decision-making exchanges sensitive electronic health records (EHRs) and personal health records (PHRs) among healthcare entities like insurance companies, cured patients, researchers, doctors, drug distributors, pharmacists, etc. Regrettably, the privacy of EHRs, sensitive

information, and PHRs are frequently compromised during medical collaboration. Interoperability issues among healthcare stakeholders carry extra barriers to effective data sharing for collaborative decision-making in the medical sector. While designing new schemes to overcome these security issues, leveraging privacy-aware blockchain technology can be a substantial option.

Accenture's [4] consumer survey reported that around 26% of consumer's electronic medical records (EMRs) in the USA were breached, where 31% of those lost social security number, contact information, EHRs, and PHRs. Guardian [5] reported the top 10 healthcare data breaching incidents and found 'Anthem Blue Cross,' a giant health insurance firm, as the number one data breaching company. The company disclosed 80 million data including sensitive medical data, social security numbers, addresses, etc. Several studies discussed the various cyber-physical risk factors in the healthcare domain [6–8]. Another study [9] discussed the risk of healthcare-related data leaking in a wireless body area network, cloud, and 5G. US-based identity theft resource center (ITRC) [10] announced businesses (55%) and healthcare industries (23.7%) as the top two identity breaching sectors from 2005 to 2017 (total 1579 breaches in the USA, 2017). False data injection attack in healthcare has been a widely discussed topic in clinical data security and the privacy domain [11].

A few reasons for clinical data breaching as reported by Snell [12] are unintended data disclosure (41%), hacking and malware (19%), internal incidents (15%), and physical damage (8%). It identified 380 dollars as a 'breaching cost' for every health care record lost. Around 63% of patients visit an online platform for medical help and 62.4% of them trust their doctor for data sharing [13]. Lack of participants, sloppy stakeholder's identity securing mechanisms, corruption in decision picking mechanisms, lack of accountability, data manipulation, delayed communication, and corrupted mediums have been recognized as the major worries of collaborative medical decision-making [14–19]. Despite aforesaid challenges, collaboration can always upgrade the existing healthcare amenity. The Health Insurance Portability and Accountability Act (HIPAA) [20] identified 'healthcare provider' (doctor and hospital) as the second largest identity leaking sector (Table 1).

**Table 1.** Responsible institutions for the healthcare data breach.

| Entity Year | Provider | Healthcare Planer | Business Associate | Other | Total |
|---|---|---|---|---|---|
| 2009 | 14 | 1 | 3 | 0 | 18 |
| 2010 | 134 | 21 | 44 | 0 | 199 |
| 2011 | 137 | 20 | 42 | 1 | 200 |
| 2012 | 155 | 22 | 36 | 4 | 217 |
| 2013 | 199 | 18 | 56 | 5 | 278 |
| 2014 | 202 | 71 | 41 | 0 | 314 |
| 2015 | 196 | 62 | 11 | 0 | 269 |
| 2016 | 257 | 51 | 19 | 0 | 237 |
| 2017 | 288 | 52 | 19 | 0 | 259 |
| Total | 1582 | 318 | 271 | 10 | 2181 |

Presently, patients collaborate (Collaborations$_{1-4}$) with entities one at a time (Times$_{A1-A5}$). This type of collaboration costs time, money, and the privacy of users (Figure 1).

Existing collaborative healthcare decision-making has the following limitations [14–19,21–23]:

- Multi-dimensional personal data privacy policies imposed by individual institutes.
- Clinical data, medical decisions, and patient identities are being manipulated and distributed without proper acknowledgment and consent.
- Scalability and management difficulties during collaborative decision gathering and storing.
- Usage of centralized architecture with a single point of failure.
- Incompatibility with institutional and regional regulations in SDM-based medical treatment.
- Lack of association among health professionals.

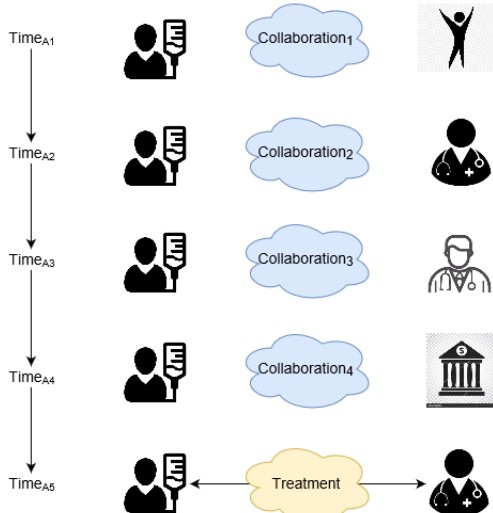

**Figure 1.** The time delay in existing collaborative medical decision making.

PII is the information that solely identifies individual identity. Similarly, information that gives full identity after combining a few of them is potential personally identifiable information (PPII). Alternatively, non-personally identifiable information (NPII) is fully anonymous. Kayaalp [24] classified protected health information (PHI) as a common set of PII and medical records (Figure 2).

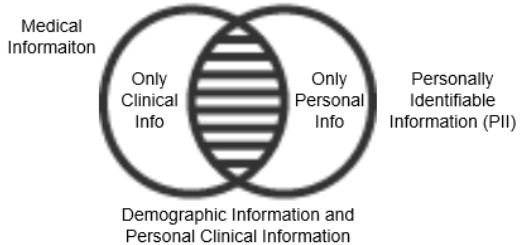

**Figure 2.** Relationship of personal information and medical information demographically.

To address the risks of shared medical decision-making, blockchain can be a rational solution. This technology has already been used beyond cryptocurrency [25]. This distributed technology can create a transparent network for healthcare stakeholders (i.e., insurance company, cured patient, doctor). Concurrent hosting of stakeholders in a single platform is also possible without a trusted third party. General Data Protection Regulation (GDPR), Health Insurance Portability and Accountability Act (HIPAA), and National Electronic Health Transition Authority (NEHTA) are conflicted with the current blockchain. Similarly, the recently activated GDPR [26] demands personal data erasure [27]. 'Right to be forgotten' of GDPR conflicts with the basic architecture of blockchain's 'data immutability'. On the contrary, privacy-preserved blockchain architectures are also available [28,29]. Chainforg [30] listed possible solutions: No storing of personal information on the blockchain, record personal data pseudo-anonymously, store information in a referenced local encrypted database. Similarly, this study proposes a health data privacy preserving blockchain scheme and associated consensus gathering algorithm, proof of familiarity (PoF).

## 1.1. The Contribution of the Paper

This study proposes a collaborative medical decision gathering a private blockchain scheme and associated proof of familiarity (PoF) consensus algorithm. PoF enables simultaneous (Time$_{A1}$) collaboration (Collaboration$_1$ to Collaboration$_2$) (Figure 3) to save time, money, and privacy of users.

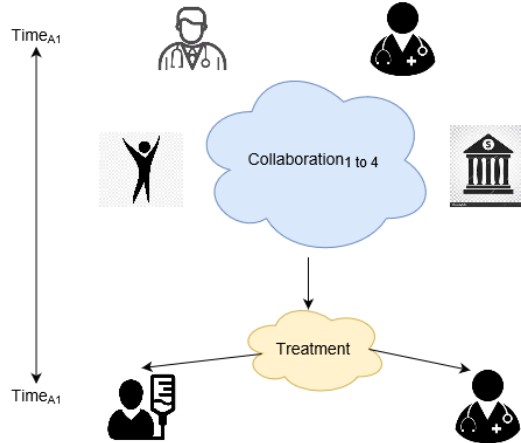

**Figure 3.** Proposed collaborative medical decision-making.

To overcome issues of the aforementioned collaborative medical decision-making, this study makes the following contributions:

- This scheme provides a seamless sharing of decisions among the stakeholders.
- This scheme stores medical decisions and identities with an off-chain blockchain to increase confidentiality. Hence, the study fully complies with existing privacy regulations regarding PII.
- PoF, the first consensus algorithm so far, reflects the doctor's skill, cured patient's feedback, insurance company's policy, and blockchain's security concurrently in decision making.

*1.2. Roadmap of the Paper*

The rest of this paper is organized as follows: Section 2 reviews existing works on blockchain and medical decision-making. Section 3 presents the proposed medical decision gathering scheme, PoF algorithm, system explanation, and a use case. Section 4 covers test detail, advantages, limitations, and evaluation with other schemes. Section 5 concludes the work and is followed by references.

## 2. Literature Review

Exchange of medical decisions helps patients to choose a medical decision proficiently [31]. A study [32] termed blockchain as 'security by design' and a 'trust machine'. This section elaborates the strength and weaknesses of blockchain in healthcare [33,34].

*2.1. Blockchain*

Nakamoto [35] introduced blockchain technology by proposing electronic cash known as bitcoin. In bitcoin [35], transactions are confirmed by cryptography without any central administrative authority. Cryptocurrencies [36] are entirely decentralized and offer anonymized transparency with the virtue of a freely accessible distributed ledger, blockchain. Key properties of blockchain technology adopted by PoF based medical decision-making system are mentioned here:

1. Trusted transaction: Blockchain can digitally authenticate transaction regulations and nodes. Finally, the necessity of a trusted third party is useless. This largely reduces the settlement cost.
2. Decentralized storage: Blockchain stores multiple copies of data in a physically parted location. Data integrity can be checked from all available nodes to remove a single point of failure.
3. Traceability: Verification of corrupted information and nodes are possible by hash and timestamp validation. Data traceability also empowers the identification of the corrupted node. Eventually, this traceability and loyalty of blockchain encourages stakeholders for collaboration.
4. Improved security: Double layer security (private-public key) checking and immutability of information with hash storing increases the security of blockchain over existing technologies.

Few frequently used jargons of blockchain technology are discussed here:

- Header: Header stores information like version, threshold, timestamp, nonce, etc.
- Transaction counter: Transaction counter reflects the current block number.
- Transaction data: Use of this field varies on purpose (i.e., decision, rating, money, privacy rule etc.).
- Consensus algorithm: Consensus algorithms are designed to validate the reliable node and agree on a suitable decision among nodes. Proof of work (PoW) [35], practical Byzantine fault tolerance (PBFT) [37], proof of stake (PoS) [38], ripple [39], tendermint [40], etc. are few of them.
- Forking: Forking happens if a transaction faces concurrent and multiple correct paths. Three kinds of forking are hard fork, soft fork, and user-centric fork [41].

Some performances assessing factors of three blockchain types are compared (Table 2). If we notice the privacy row (Table 2), private and consortium blockchain carry higher personal data privacy because transactions are private to the nodes within the membership [41]. Personal information is shared among trusted private nodes only. In case of any blockchain failure, personal information is leaked within the group. So, PoF finds a valid reason to use cost-effective consortium blockchain.

**Table 2.** Types of blockchain (public, private and consortiums).

| Type \ Facts | Public Blockchain | Private Blockchain | Consortium Blockchain |
|---|---|---|---|
| Definition | All participant can read, write, monitor, transact, download and start running a public node. | Instead of allowing anyone to control and participate in the verification, some specific number of nodes are selected. | Owner of the blockchain has the right to make rules and other terminologies. |
| Cost | High | Medium | Low |
| Security | High | Medium | Medium |
| Privacy | Medium | High | High |
| Transaction rate | Slower | Faster | Fastest |
| Architecture | Decentralized | Partially decentralized | Partially decentralized or centralized |
| Example | Bitcoin, Ethereum, Litcoin | R3, Corda, B3i | Monax, Multichain |
| Use sector | Cryptocurrency | Government | Organization |

PwC [42] estimated almost 77% of surveyed companies will introduce blockchain technology by 2020. Some use cases of blockchain technology include finance [43], internet of things [44], big data [45], human resources [46], supply chain [47], booking and registration [48], personal data protection [49], and education [50]. Swan [51] compared this technology as a smart Hyperledger that stores exchanged information as an immutable new block after an independent consent from all attendees. Transaction decisions are approved after a secure consensus algorithm to make blockchain a decent candidate for the fourth industrial revolution [52]. Studies [45,53,54] have also used blockchain as a medium of secure decision gathering. Pinheiro [53] used blockchain to develop a multi-agent system that gains knowledge from a network during decision making. Beside financial deal, blockchain technology can also manage decisions, instructions, reviews, and message [45].

### 2.2. Collaborative Medical Decision-Making and Blockchain

Healthcare researchers and policymakers increasingly emphasize SDM for quality treatment [31]. Collaborative medical decision-making has been a common practice in healthcare industries [14–19,21,22,55]. Schwartz [56] defines collaborative medical decision-making as:

*Medical decision science is a field that encompasses several related pursuits. As a normative endeavor, it proposes standards for ideal decision making. As a descriptive endeavor, it seeks to explain how physicians and patients routinely make decisions, and has identified both barriers to, and facilitators of, effective decision making. As a prescriptive endeavor, it seeks to develop tools that can guide physicians, their patients, and health care policymakers to make good decisions in practice.* [56]

Medical billing, distance patient monitoring, and clinical data exchanging are a few use cases of blockchain technology in the healthcare sector. Medical decision-making, insurance policy sharing, clinical data sharing, and preserving are also managed by blockchain [16,33,34,57–60].

Graffigna [16] considered the patient's engagement in healthcare as a key factor to improve the overall medical decision. The study introduced the patient health engagement (PHE) model that includes paternalistic decision-making, informed decision-making, and shared decision-making. This study encourages the active participation of patients in collaborative decision-making. Similar encouragement was also provided by Jayanti [57] who considered clinical collaboration and payors decisions for upgrading the clinical verdict. The study [57] enrolled 458 patients to find experience (age) with particular diseases as a key aspect in collaborative medical decision-making. At the same time, the study considered the experience and achievement of the decision giving entities (doctor, patient, insurance). So, our study ensures both patients and doctors get an equal participation opportunity in the PoF based collaborative medical decision-making system.

A secure medical data sharing platform called 'medblock' was proposed by Kai Fan [61]. The study claimed privacy awareness during EMR sharing. The study stored the exact location of data encryption to track EMR access and retrieve. Although the study [61] ensured higher security concept, it failed to provide enough privacy of the patient's identity. Similarly, the study [61] claimed energy efficiency in 'medblock' but failed to prove any strong reasons behind their claim.

Another study [62] proposed an architecture that stores every possible transaction information, patients identities, medical decisions, insurance records, etc. in a single block. Similarly, Seo-Joon [63] introduced Blockchain Applied FASTQ and FASTA Lossless Compression (BAQALC), an efficient blockchain-based platform for DNA sequence transmission. This study also suggests to store the compressed DNA sequence in a block. The main drawback of these studies is that they overlooked the average block size. Oppositely, the proposed system follows a separate storing of personal and non-personal information. The local storing of personal data reduces the overall block size in the blockchain.

Azaria [64] introduced another medical data handling and decision achieving platform supported by blockchain technology. The study proposed 'MedRec', a platform that works on PoW to manage EHR transparently. The study [64] rewarded miners unwanted access to EHR. Therefore, the study seriously hampered personal information privacy. In comparison to that, the proposed PoF protects the complete identity of the participants by restricting the access of third parties.

Kuo [33] proposed a 'modelchain' that used machine learning and blockchain technology to simplify patient-centered outcomes research (PCOR). In one side, the study hid node identity and on the other side proposed inter-institution collaboration with a consortium blockchain. This architecture decreased the trust of the overall system many times. On the contrary, this study (PoF) exposes user identity to the consortium chain. This ensures easy identification of the corrupted node.

Ozercan [34] proposed gene token as a medium of gene information exchange. An off-the-chain and on-the-chain gene data storing platform were proposed by the study [34] that provided a scheme for sharing biological data among third parties.

Shae [59] proposed privacy aware health data management with blockchain technology. The study proposed four separate layers under clinical trial. They were: (a) A distributed paradigm with blockchain technology; (b) application information management; (c) secure and anonymous identity management; (d) privacy-preserved trusted information management. However, the study failed to adopt standard blockchain architecture. That is, the lack of coherence among presented components reduced the large-scale implementation scope. On the other hand, PoF based complies with both standard blockchain architecture and existing privacy standards.

Zhang [28] proposed a 'fast healthcare interoperability resources (FHIR) chain' for collaborative clinical decision-making. The study identified four key challenges of collaborative medical decision-making in healthcare. Security concern, lack of trust, scalability concern, and lack of exchanged data standard are the barriers of collaborative clinical decision-making. The PoF based study successfully responded to all the aforementioned issues to propose an efficient collaboration scheme.

Alevtina [65] proposed EMRs sharing blockchain platform in collaboration with Stony Brook University Hospital. The study used oncology-specific data aggregation from patients and doctors. This study also uses off-chain data storage. However, no concrete consensus gathering was proposed.

Alex [66] and Omar [60] analyzed the performance of blockchain-based PHI exchange platforms that preserve privacy. Another study [67] proposed a hierarchy based blockchain wallet called 'Youbase'. As 'Youbase' stores personal information in a tree-like structure with keys (public and private), it can store information separately regardless of data type. Personal data are anonymized to comply with the GDPR. However, consensus gathering time is an issue of this study.

Another study [68] had also mentioned that the transaction rate and waiting time may doubt the large-scale use of blockchain. On the contrary, the proposed study successfully addresses those issues by reducing the block size and increasing the transaction rate. Another study [55] explained the need for a new blockchain architecture to overcome existing privacy issues.

As aforementioned studies have already questioned the integrity of stakeholders, blockchain, with no central authority, is a possible solution to bring trust to medical collaboration. This study, therefore, finds a valid cause to use blockchain in medical decision-making in support with existing studies previously discussed. Table 3 compares the proposed study with a few other studies [28,61,65,69,70]. During comparison in Table 3 key contributions of the proposed PoF based medical decision-making system are considered.

**Table 3.** Comparison of blockchain-based medical and healthcare architecture.

| Features \ Schemes | Chen [69] | Fan [61] | Zhang [28] | Tseng [70] | Ryu [65] | Proof of Familiarity |
|---|---|---|---|---|---|---|
| Blockchain architecture | Public | Public | Public | Private | Public | Consortium |
| Off-chain data | No | No | No | No | No | Yes |
| Proposed new consensus | No | Yes | Yes | No | No | New |
| Implementation cost | High | High | High | Medium | Low | Low |
| Purpose of use | Data | Data | No | Drug supply | Data | Decision-making |

## 3. Proposed Method

This section discusses the consortium blockchain-based critical medical decision achieving scheme followed by a proposed proof of familiarity (PoF) consensus gathering algorithm.

### 3.1. Architecture and Flow Chart of the Whole System

Figure 4 describes the interaction among medical entities (cured patient, doctor, insurance company) of the proposed blockchain-based collaborative medical decision gathering scheme. If a patient needs collaboration, a request is made to the cured patient, doctor, and insurance company through PoF. Then, cured patients return past treatment experience, doctors return their medical verdict, and insurance companies return their policy back to the PoF. After calculation, PoF returns a successful collaborative medical decision to the patient. During the process, a successful collaborative medical decision and decision providing stakeholder's identities are also recorded. The PII of participating entities is stored in associated local databases. The collaborative medical decision and the hash of locally stored PII are stored in the blockchain (Figure 4). Four major entities are expressed as patient, P; cured patient $C_p$; doctor, D; and insurance company, $I_c$ (Figures 4 and 5). All major entities hold a local database ($DB_{loc}$) and connected blockchain database ($DB_{bc}$). Finally, an application programming interface (API) poses by four major entities as a communication interface (Figures 4 and 5).

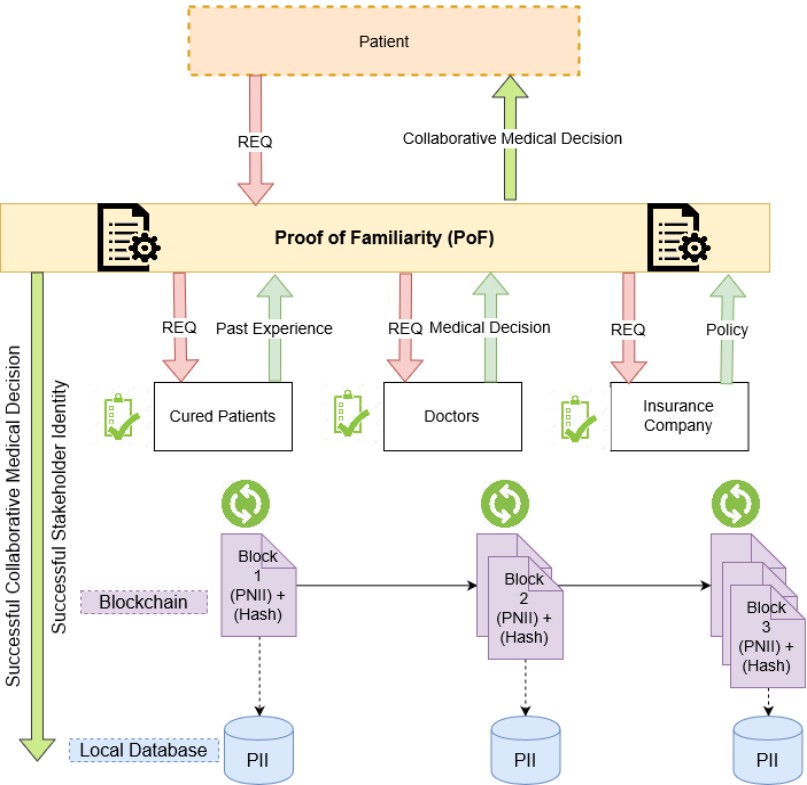

**Figure 4.** The system model of the collaborative medical decision gathering scheme (proof-of-familiarity (PoF) consensus).

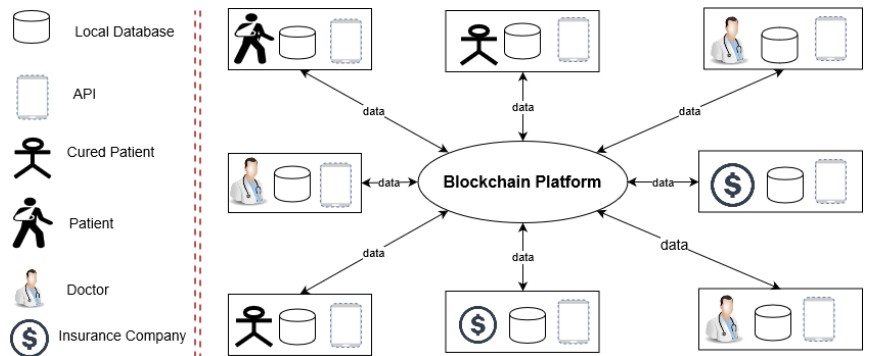

**Figure 5.** Stakeholders (participating entities) collaborating for medical decision gathering in PoF.

To adopt core blockchain architecture, the major entities (P, D, $I_c$, and $C_p$) are considered as a node. In PoF based scheme, all nodes are assumed to be located within a consortium blockchain network and connected via the internet. Node structure and functional properties are discussed here:

- Patient (P): Patient, P might be deprived and located in a distressed area without ample medical facilities. By this scheme, P achieves a decent treatment (medical decision) by collaborating with D, $I_c$, and $C_p$. P is equipped with a $DB_{loc}$, an API, and a $DB_{bc}$ to manage identities and decisions.
- Doctor (D): Doctor, D collaborates with D, $I_c$, and $C_p$ through PoF to help P with an expert decision. D has a $DB_{loc}$, an API, and a $DB_{bc}$ to manage identities and decisions.
- Cured patient ($C_p$): Cured patient, $C_p$ represents patients already treated with a similar disease. $C_p$ collaborates with D, $I_c$, and $C_p$ and helps P with the experience $C_p$ gathered during diagnosis. A $C_p$ also uses a $DB_{loc}$, an API, and a $DB_{bc}$ to manage identities and decisions.

- Insurance company ($I_c$): Insurance company, $I_c$ deals with health-related financial strategies of a P. $I_c$ uses a $DB_{bc}$, an API, and a $DB_{loc}$ to manage identities and decisions.

Data categorization and storing: PII, PPII, and NPII are data types and $DB_{bc}$ and $DB_{loc}$ are database types used in PoF based scheme. Already discussed GDPR [26,27] demands the 'right to be forgotten' or 'data erasure' of personal data (PII and PPII). So, the proposed system stores PII and PPII in an erasable $DB_{loc}$ of P, D, $I_c$, and $C_p$. On the contrary, NPII and the hash of locally stored PII and PPII are stored in an immutable $DB_{bc}$. Doctor's identification information, patient's identification information, cured patient's identification information, patient's sensitive clinical data, and insurance companies' identification information are stored in a $DB_{loc}$. Oppositely, non-identifiable profile information, smart contract, clinical metadata, the hash of a local database, and non-identical decisions are stored on a $DB_{bc}$.

### 3.2. Proposed Proof of Familiarity (PoF) Consensus Algorithm

#### 3.2.1. Functional Architecture Proof of Familiarity (PoF)

In PoF, decisions are taken based on the individual's level of expertise, experience, and success rate. We firstly posit that D, $I_c$, and $C_p$ involved in a particular occupation or experience for a longer period also have a better decision-making ability. We secondly posit that D, $I_c$, and $C_p$ already provided a correct decision, also have a higher chance of taking the correct decision in future collaboration. We explore the above two conjectures to propose that if D, $I_c$, and $C_p$ provide an opinion or medical decision, the winner is always selected based on the previous two propositions. To express individual qualitative achievements numerically, individual familiarity index (IFI) is calculated. IFI starts at 0.0 and can be up to 1.0. IFI is updated according to the performance history of D, $I_c$, and $C_p$ stored in $DB_{loc}$ and $DB_{bc}$. IFI index of PoF depends on several considering factors:

- The judgement of the doctor: While choosing a doctor's decision, factors like job experience time ($J_{ET}$) and treatment success rate ($T_{SR}$) are considered. IFI of a doctor (IFI_D) is calculated as IFI_D = {$J_{ET}$, $T_{SR}$}.
- The perspective of the cured patient: Proposed system considers factors like treatment experience time ($T_{ET}$), current condition ($C_C$), and experience of disease ($E_D$) of cured patients. IFI of a cured patient (IFI_$C_p$) is calculated as: IFI_$C_p$ = {$T_{ET}$, $C_C$, $E_D$}.
- Perception of insurance company: Financial policy is another significant aspect of collaborative medical decision-making. IFI of an insurance company (IFI_$I_c$) is calculated from settlement time ($S_T$) and cover amount ($C_A$). IFI_$I_c$ is calculated as: IFI_$I_c$ = {$S_T$, $C_A$}.
- Final collaborative medical decision: Finally, PoF calculates the final collaborative medical decision, $C_d$. Based on the IFI value (IFI_$I_c$, IFI_$C_p$, and IFI_D) of three entities, a winner is chosen from each of the entities (D, $I_c$, and $C_p$). After combining three winner's decisions, $C_d$ is decided for P.

#### 3.2.2. Transactions in Proof of Familiarity (PoF)

This section shows interactions of PoF and P, D, $I_c$, $C_p$, $DB_{loc}$, $DB_{bc}$ to finalize $C_d$ for P (Figure 6).

#### 3.2.3. Block Architecture

Architecturally, all blocks are connected to a previous block (block 0 to block N). A block stores a decision and identities after a PoF approval. PoF does not allow any mining award or transaction fee. Since the proposed blockchain stores identities of P, $C_p$, D, and $I_c$ in a well-secured manner, no nonce is used in PoF. The only incentive D, $I_c$, and $C_p$ achieve is uprise of IFI if the patient receives a successful decision through PoF. Block 1 (Figure 7) is described here:

- Hash of last block: Hash of recently created block is stored by this field. This tracks originality of the data of this block. If modification happens to block 0, the header of block 1 will no

longer point to the header of block 0. This corrupted block is disconnected from the rest of the block (blockchain).

- Timestamp: This field stores the approval time of the final collaborative medical decision, $C_d$.
- Flag 1: Patient's NPII and clinical history are stored in 'Flag 1' field.
- Flag 2: Doctor's NPII and decision history are stored in 'Flag 2' field.
- Flag 3: Insurance company's NPII and policy history are stored in 'Flag 3' field.
- Flag 4: Cured patient's NPII and clinical history are stored in 'Flag 3' field.
- PII Hash: Hash of the locally stored PII is stored in 'hash' field. This can cross check the originality of the information stored in local databases of P, D, $I_c$, and $C_p$.
- Decision: Accepted final collaborative medical decision, $C_d$ are stored in 'decision' field.
- NPII: All other significant NPII (rules, conditions) is stored in the 'NPII' field of the block.

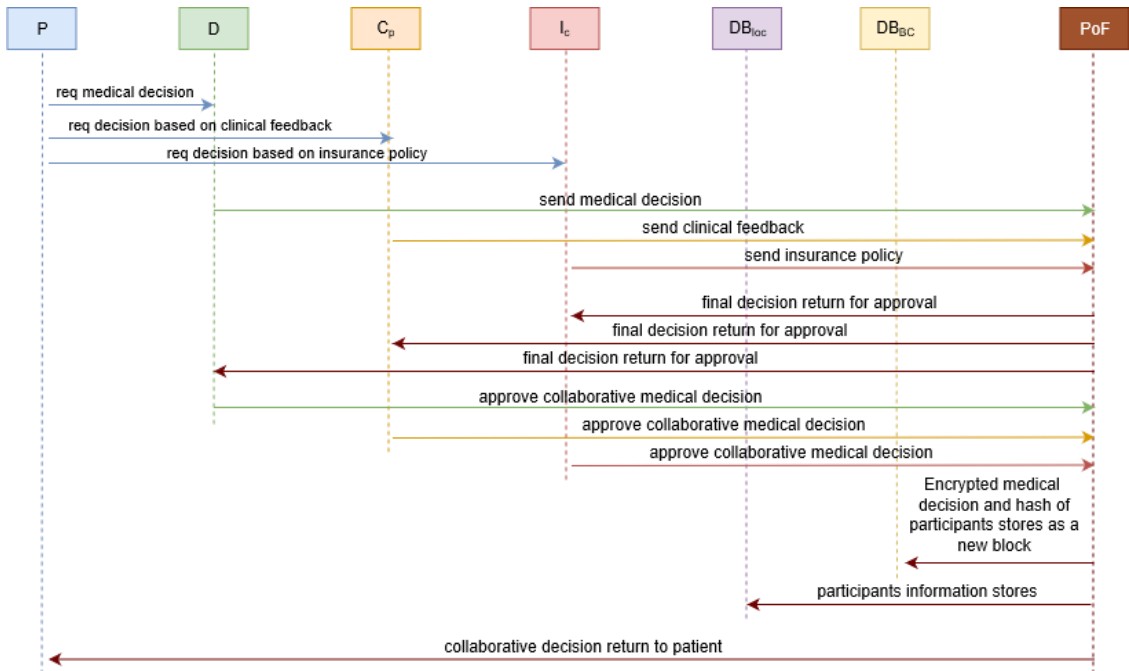

**Figure 6.** Transaction among stakeholders in proposed PoF.

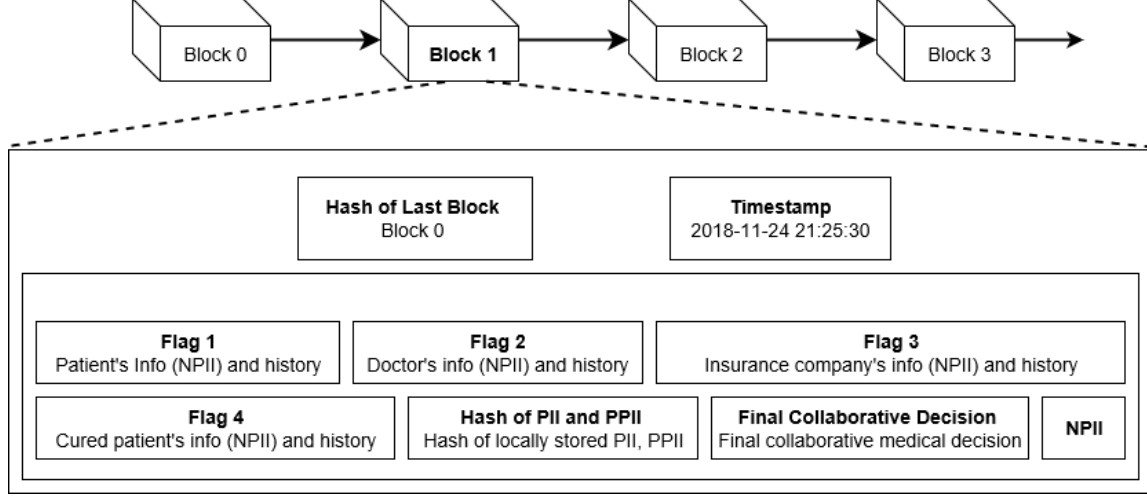

**Figure 7.** Sample block architecture of the proposed scheme (Block 1).

### 3.2.4. Proof of Familiarity (PoF) Algorithm

This section elaborates PoF in detail that works mostly on the efficiency of participated entities on several affecting factors mentioned in Section 3.2.1. PoF consensus gathering algorithm is divided into two sub-algorithms. Firstly, the selection of final participating entities is done by Algorithm 1. Secondly, the creation and storing of the final decision are done by Algorithm 2. At this point, two algorithms are proposed followed by a step by step analysis with adequate elaboration.

---

**Algorithm 1:** Selection of final participating entities

---

**Input**:

patients, P; cured patients, $C_p$; doctors, D; insurance companies, $I_c$; // all participating entities

individual decision, $I_d$; // decision of participating entities

treatment experience time, $T_{ET}$; current condition, $C_C$; experience of disease, $E_D$ // decision judging factors of cured patients

job experience time, $J_{ET}$; treatment success rate, $T_{SR}$; // decision judging factors of doctors

settlement time, $S_T$; cover amount, $C_A$ // decision judging factors of insurance companies

blockchain database, $DB_{bc}$; local database, $DB_{loc}$ // storing entities (local and blockchain database)

**Output**:

//individual familiarity index (IFI) of participating entities

individual familiarity index of cured patient, IFI_$C_p$; Individual familiarity index of doctor, IFI_D; individual familiarity index of insurance company, IFI_$I_c$

finally selected cured patient, $C_p$_FS; finally selected doctor, $D_{FS}$; finally selected insurance company, $I_c$_FS

individual familiarity index of finally selected cured patient, IFI$_{FS}$_$C_p$; individual familiarity index of finally selected doctor, IFI$_{FS}$_D; Individual familiarity index of finally selected insurance company, IFI$_{FS}$_$I_c$

individual decision of finally selected cured patient, $I_{d\_FS\_Cp}$; individual decision of finally selected doctors, $I_{d\_FS\_D}$; individual decision of finally selected insurance company, $I_{d\_FS\_Ic}$

**Begin**:

verification of $C_p$, D, $I_c$ from $DB_{loc}$ and $DB_{bc}$ // identity of participating entities are verified

verification of $J_{ET}$, $T_{SR}$, $T_{ET}$, $C_C$, $E_D$, $S_T$, $C_A$ from $DB_{loc}$ and $DB_{bc}$ // decision judging factors are verified

verification of $I_d$ of $C_p$, D, $I_c$ from $DB_{loc}$ and $DB_{bc}$ // individual decisions are verified

for each ($C_p$, D, $I_c$)

IFI_$C_p$ = Sum ($T_{ET}$, $C_C$, $E_D$) // individual familiarity index of cured patients

FI_D = Sum ($J_{ET}$, $T_{SR}$) // individual familiarity index of doctors

IFI_$I_c$ = Sum ($S_T$, $C_A$) // individual familiarity index of insurance companies

end

for each ($C_p$, D, $I_c$)

$C_p$_Fs = $C_p$ with max (IFI_$C_p$) // finally selected cured patient is chosen with maximum IFI

$D_{FS}$ = D with max (IFI_D) // finally selected doctor is chosen with maximum IFI

$I_c$_FS = $I_c$ with max (IFI_$I_c$) // finally selected insurance company is chosen with maximum IFI

end

for each ($C_p$, D, $I_c$)

$I_{d\_FS\_Cp}$ = $I_d$ with max (IFI_$C_p$) // individual decision of finally selected cured patient

$I_{d\_FS\_D}$ = $I_d$ with max (IFI_D) // individual decision of finally selected doctor

$I_{d\_FS\_Ic}$ = $I_d$ with max (IFI_$I_c$) // individual decision of finally selected insurance company

end

for each ($C_p$, D, $I_c$)

IFI$_{FS}$_$C_p$ = $I_d$ with max (IFI_$C_p$) // IFI of finally selected cured patient

IFI$_{FS}$_D = $I_d$ with max (IFI_D) // IFI of finally selected doctor

IFI$_{FS}$_$I_c$ = $I_d$ with max (IFI_$I_c$) // IFI of finally selected insurance company

end

**End**

---

Selection of final participating entities: All participating entities (P, $C_p$, D, $I_c$) are verified from $DB_{loc}$ and $DB_{bc}$. After that, associated decision judging factors like: Job experience time ($J_{ET}$), treatment

success rate ($T_{SR}$); treatment experience time ($T_{ET}$), current condition ($C_C$), experience of disease ($E_D$), settlement time ($S_T$), and cover amount ($C_A$) are verified from $DB_{loc}$ and $DB_{bc}$. Individual decision, $I_d$ are verified from $DB_{loc}$ and $DB_{bc}$ as well. Individual familiarity index (IFI) of participating entities (IFI_$C_p$, IFI_D, IFI_$I_c$) are calculated by summing up entity wise decision judging factors ($J_{ET}$, $T_{SR}$, $T_{ET}$, $C_C$, $E_D$, $S_T$, $C_A$). Out of all participating entities ($C_p$, D, $I_c$), entities with maximum individual familiarity (IFI_$C_p$, IFI_D, IFI_$I_c$) are selected as finally selected participating entities ($C_{p\_FS}$, $D_{FS}$, $I_{c\_FS}$). At this stage, individual decision of finally selected participating entities ($I_{d\_FS\_Cp}$, $I_{d\_FS\_D}$, $I_{d\_FS\_Ic}$) with maximum individual familiarity index (IFI_$C_p$, IFI_D, IFI_$I_c$) are chosen. Individual familiarity index of finally selected participating entities (IFI$_{FS}$_$C_p$, IFI$_{FS}$_D, IFI$_{FS}$_$I_c$) with maximum individual familiarity index (IFI_$C_p$, IFI_D, IFI_$I_c$) are chosen as well in the first part of the PoF algorithm.

---

**Algorithm 2:** Selecting and storing of final collaborative medical decision

---

**Input**:

finally selected cured patient, $C_{p\_FS}$; finally selected doctor, $D_{FS}$; finally selected insurance company, $I_{c\_FS}$
individual familiarity index of finally selected cured patient, IFI$_{FS}$_$C_p$; individual familiarity index of finally selected doctor, IFI$_{FS}$_D; individual familiarity index of finally selected insurance company, IFI$_{FS}$_$I_c$
individual decision of finally selected cured patient, $I_{d\_FS\_Cp}$; individual decision of finally selected doctors, $I_{d\_FS\_D}$; individual decision of finally selected insurance company, $I_{d\_FS\_Ic}$
blockchain database, $DB_{bc}$; local database, $DB_{loc}$ // storing entities (local and blockchain database)

**Output**:

final collaborative medical decision, $C_d$

**Begin**:

$C_{p\_FS}$, $D_{FS}$, $I_{c\_FS}$ broadcast $I_{d\_FS\_Cp}$, $I_{d\_FS\_D}$, $I_{d\_FS\_Ic}$ to Cp, D, Ic //decision broadcast for consensus
$C_d$ = combination of ($I_{d\_FS\_Cp}$, $I_{d\_FS\_D}$, $I_{d\_FS\_Ic}$) //aggregation of individual decision in medical decision
$C_d$ stores in $DB_{bc}$ //final collaborative medical decision is stored in blockchain
$C_{p\_FS}$; $D_{FS}$; $I_{c\_FS}$ stores in $DB_{loc}$ //finally selected entities' identities are stored in local database
Hash of locally stored $C_{p\_FS}$, $D_{FS}$, $I_{c\_FS}$ from $DB_{loc}$ stores in $DB_{bc}$ // hash are stored in blockchain
IFI$_{FS}$_$C_p$, IFI$_{FS}$_D, IFI$_{FS}$_$I_c$ stores in $DB_{bc}$ // IFI of finally selected participants are stored in blockchain
$J_{ET}$, $T_{SR}$, $T_{ET}$, $C_C$, $E_D$, $S_T$, $C_A$ of $C_{p\_SPE}$, $D_{SPE}$, $I_{c\_SPE}$ is updated in $DB_{bc}$ and $DB_{loc}$

**End**

---

Selecting and storing of final collaborative medical decision: Finally selected participating entities ($C_{p\_FS}$, $D_{FS}$, $I_{c\_FS}$) broadcast individual decisions ($I_{d\_FS\_Cp}$, $I_{d\_FS\_D}$, $I_{d\_FS\_Ic}$) to all other $C_p$, D, $I_c$ for consensus gathering. Afterwards, final collaborative medical decision ($C_d$) is made by combining individual decisions ($I_{d\_FS\_Cp}$, $I_{d\_FS\_D}$, $I_{d\_FS\_Ic}$) followed by storing in $DB_{bc}$. Identities of selected participating entities ($C_{p\_FS}$, $D_{FS}$, $I_{c\_FS}$) are stored in their individual $DB_{loc}$. Hash of locally stored identities ($C_{p\_FS}$; $D_{FS}$; $I_{c\_FS}$) is also stored in $DB_{bc}$. Finally, individual decision judging factors ($J_{ET}$, $T_{SR}$, $T_{ET}$, $C_C$, $E_D$, $S_T$, $C_A$) of successful entities ($C_{p\_SPE}$, $D_{SPE}$, $I_{c\_SPE}$) are updated for future use.

## 3.3. Running Example

This section represents IFI and considers factors of 3 doctors (D), 2 cured patients ($C_p$), and 2 insurance companies ($I_c$) with a running example from time 0 to time 4 in Table 4 (Figures 8–12). Values stated in Table 4 are arbitrarily considered, where actual values will be calculated based on Section 3.2.1, Algorithm 1, and Algorithm 2.

- At time 0: Patient, $P_1$ broadcasts desired medical problem to $D_1$, $D_2$, $D_3$, $C_{p1}$, $C_{p2}$, $C_{p3}$, $I_{c1}$, $I_{c2}$ for collaborative medical decision gathering (Figure 8).
- At time 1: Participating entities ($D_1$, $D_2$, $D_3$, $C_{p1}$, $C_{p2}$, $C_{p3}$, $I_{c1}$, $I_{c2}$) return individual decisions (judgment and policy) to PoF (Figure 9).
- At time 2: Based on IFI of $D_1$, $D_2$, $D_3$, $C_{p1}$, $C_{p2}$, $C_{p3}$, $I_{c1}$, $I_{c2}$ (Table 4), PoF decides decision giving entities ($D_3$, $I_{c1}$, $C_{p2}$) with maximum IFI of 0.83, 0.85, and 0.90, respectively. IFI is calculated from

the considering factors mentioned in rightmost column of Table 4. The final decisions of $D_3$, $I_{c1}$, $C_{p2}$ are sent again to $D_1$, $D_2$, $D_3$, $C_{p1}$, $C_{p2}$, $C_{p3}$, $I_{c1}$, $I_{c2}$ for final consensus (Figure 10).

- At time 3: Sensitive personal information (PII & PPII) is stored in $DB_{loc}$ of $D_1$, $D_2$, $D_3$, $C_{p1}$, $C_{p2}$, $C_{p3}$, $I_{c1}$, $I_{c2}$. Hash of the locally stored PII, PPII, and NPII are also stored as a new block (Figure 11).
- At time 4: Final collaborative medical decision, $C_d$ is created by aggregating decisions of $D_3$, $I_{c1}$, $C_{p2}$ and back to P. Lastly, IFI of successful decision-makers are raised in $DB_{loc}$ and $DB_{bc}$ for future use (Figure 12). This rise of IFI is the only incentive in the proposed PoF scheme.

**Table 4.** Individual familiarity indexes (IFI) used in running example.

| Entities | Individual Familiarity Index (IFI) | Considering Factors for Individual Familiarity Index (IFI) Calculation |
|---|---|---|
| doctor 1 ($D_1$) | 0.25 | job experience time (0.10), treatment success rate (0.15) |
| doctor 2 ($D_2$) | 0.50 | job experience time (0.30), treatment success rate (0.20) |
| doctor 3 ($D_3$) | 0.85 | job experience time (0.35), treatment success rate (0.50) |
| cured patient 1 ($C_{p1}$) | 0.45 | treatment experience time (0.20), current condition (0.10), experience of disease (0.15) |
| cured patient 2 ($C_{p2}$) | 0.90 | treatment experience time (0.30), current condition (0.30), experience of disease (0.30) |
| cured patient 3 ($C_{p3}$) | 0.60 | treatment experience time (0.30), current condition (0.25), acquaintance with disease (0.05) |
| insurance company 1 ($I_{c1}$) | 0.85 | settlement time (0.60), cover amount (0.25) |
| insurance company 2 ($I_{c1}$) | 0.50 | settlement time (0.20), cover amount (0.30) |

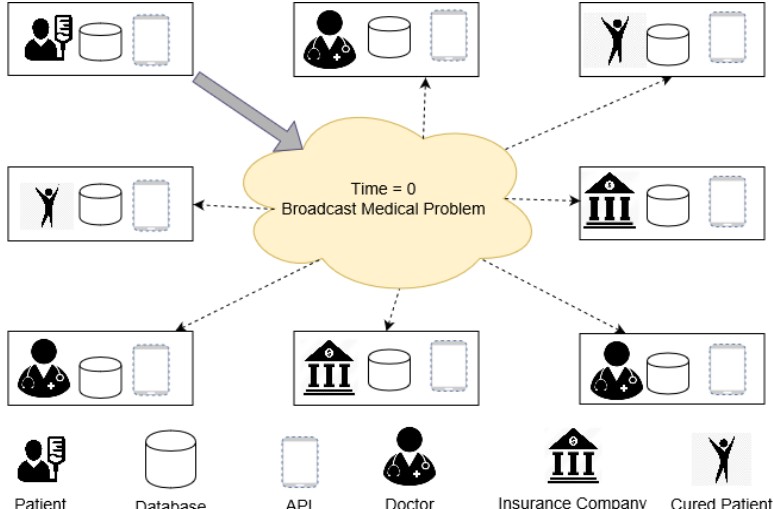

**Figure 8.** At time 0: Patient broadcasts medical problem to other entities.

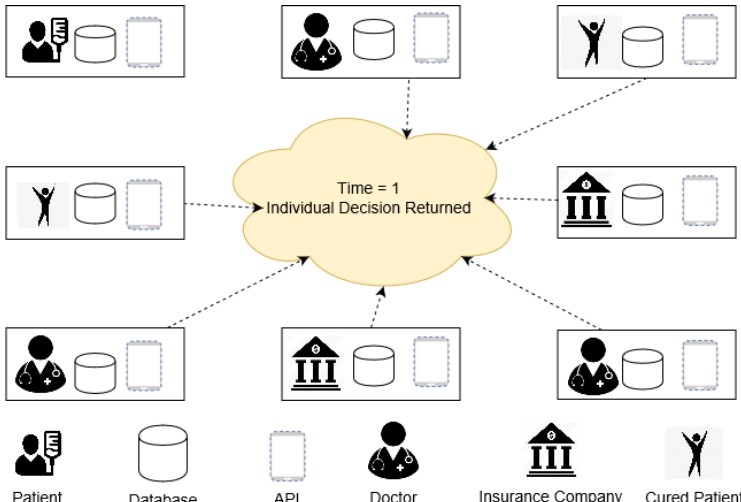

**Figure 9.** At time 1: Individual decision returns to PoF.

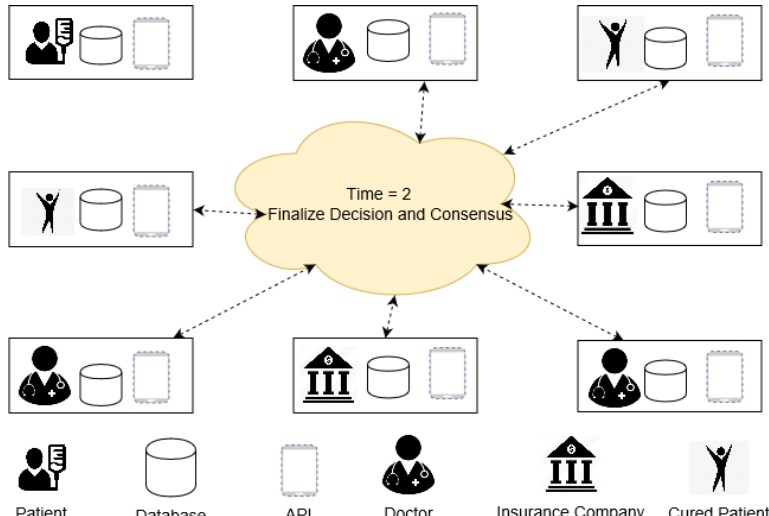

**Figure 10.** At time 2: Finalize the individual decision and consensus gathering from entities.

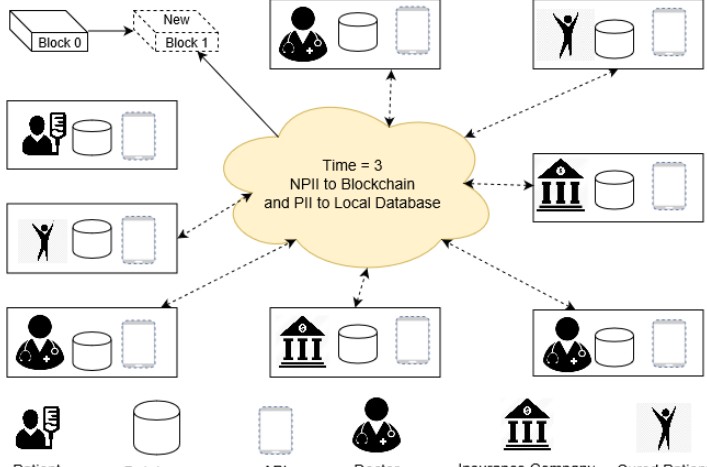

**Figure 11.** At time 3: Information storing in local and blockchain (new block created) databases.

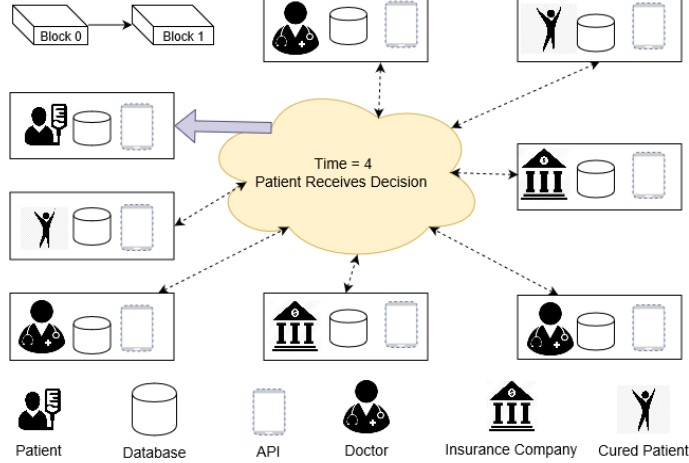

**Figure 12.** At time 4: The patient receives the final collaborative medical decision.

## 4. Implementation and Discussion

### 4.1. System Implementation

MultiChain 2.0 (alpha 3) [71,72], an open source blockchain implementation platform, was used to develop the prototype for assessing the scalability of the PoF consensus algorithm. The study prepared a consortium blockchain scenario with ten Windows 8 computers (Intel i7-6700, 3.40 GHz with 8 GB RAM Samsung Inc. Seoul, South Korea), each representing a participating entity of Table 4. This test trials a similar concept of the running example of Section 3.3 (Figure 13). The reason for choosing MultiChain 2.0 is its global acknowledgement that facilitates off-chain blockchain implementation [73–75]. Multichain 2.0 uses both blockchain and local storage. Unlike any other blockchain architectures, MultiChain 2.0 solves the problems of mining, privacy, and openness via integrated management of user permissions, thereby three folding the core aims:

- Individual node writes data in local storage.
- Hash of the data and data size are published to the network.
- After agreement from the network (consensus), the hash of locally stored off-chain data and insensitive information are stored in blockchain as a new block without any associated costs.
- Addition of a new block finishes the blockchain transaction procedure.
- To ensure privacy blockchain's activities are visible to chosen participants only.
- To enable mining to take place securely without proof of work and its associated costs.
- Facilitates privacy preservation and control over data with an easy to configure and deploy package by supporting both UNIX and Windows system.
- Provides a comprehensive JSON-RPC (Remote Procedure Call) application programming interface (API) for easy integration with the systems.
- The maximum block size is under the participant's control, it solves block size monitoring issues.
- As this is a closed (consortium) system, it considers decision only from interested participants.
- In MultiChain 2.0, users are permitted to configure configuration parameters.
- This study uses a round robin scheduling scheme for PoF. P, D, $I_c$, and $C_p$ must approve or deny a decision during their turn to generate a valid new block. For robustness of the system, mining diversity is set to 0.75 out of 1. For approving a new block, at least 75% of the total nodes or P, D, $I_c$, and $C_p$ of a block must agree and respond. This mechanism saves the system from freezing up if any set of miners become inactive for a prolonged period (Figure 13).

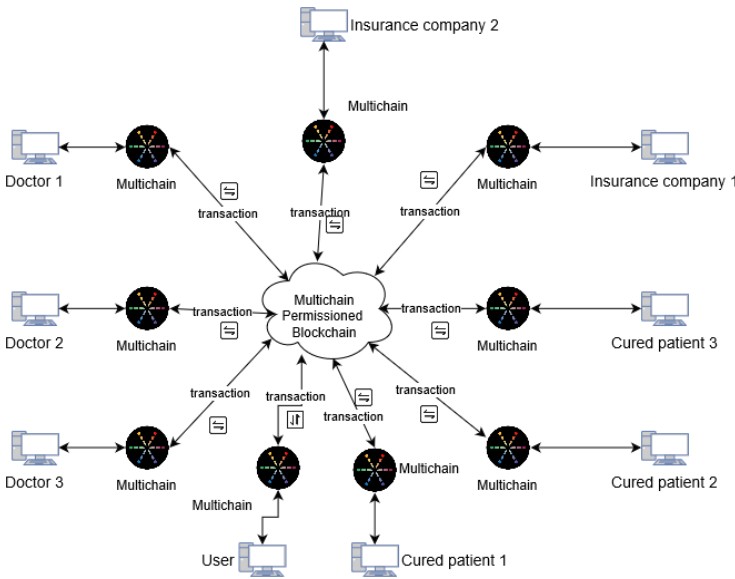

**Figure 13.** PoF communication (transaction) environment tested in the Multichain 2.0 environment.

### 4.2. Performance Analysis

#### 4.2.1. No Single Point of Failure

PoF based medical decision achieving scheme is distributed in nature while a centralized system is used by other schemes [2,3]. P, D, $I_c$, and $C_p$ also have full control over their decisions, sensitive information (PII, PPII), and identities. Blockchain stores data securely in multiple physical locations, thus no single point of failure presents. A single node cannot modify data independently to hinder overall integrity. The validity of the exchanged data and associated entities are rightly preserved.

#### 4.2.2. Smaller Block Size

One of the main concerns of blockchain technology is the large block size [68]. Since PoF stores partial information and the hash of locally stored sensitive information (PII and PPII) in a block, the use of local databases and block memory are reduced.

#### 4.2.3. Comply with Privacy Policy

Proposed system stores PII and PPII in local databases. Additionally, the hash of locally stored data (PII and PPII) and NPII are stored in the block (blockchain). Local databases of P, D, $I_c$, and $C_p$ can delete PII and PPII at any time. As leftover hashes stored in $DB_{loc}$ do not contain any sensitive data and are of no use, proposed PoF based scheme complies GDPR (right to forget and right to erase) [27].

#### 4.2.4. Security

Since medical decisions and participant's identities are stored in the blockchain, data breaching is almost impossible [32]. P, D, $I_c$, and $C_p$ never share their private keys. Similarly, identifying those secret keys (private key) are also difficult. A hacker must guess a 256-bit sequence number, in other words, $2^{256}$ probable numbers to achieve that secret key. Hashes of the locally stored PII and PPII are stored in the blockchain. Those immutable hashes inside blocks are used for verification. As a result, change in blockchain or even local databases is easily tracked to retrieve the originality of the data [32]. In PoF every data stored in the cloud (blockchain) are encrypted with private-public keys. Those keys are accessible only by the patients. In addition to that, before data are stored in the cloud-based blockchain, hashes are stored in local databases. At the same time, a digital signature by a decision giving entity also ensures data integrity.

#### 4.2.5. Cost

Distributed data storing, verification, and transaction increase the use of internet and power (ethernet and WiFi) [76]. Alternatively, PoF is comparatively cost-effective. Moreover, the cost of PoF is CAPEX (capital expenditure), whereas it is OPEX (operational expenditure) for the existing system [77]. Initial system setup covers most of the expenditure of PoF, whereas other studies [37,39,40] cost both during setup and use. Additionally, PoF saves extra cost caused for mishandling of EHRs [10,11].

#### 4.2.6. Quality of Treatment

As the proposed scheme allows patients to seek opinions from several healthcare stakeholders before making a medical decision, this upgrades the overall standard of treatment many times.

#### 4.2.7. Time

On average, existing systems spend several months to make an accurate medical decision. Oppositely, the proposed scheme collects opinion and creates a block from several parties within hours.

### 4.2.8. Confidentiality and Privacy

The PoF is designed in such a way that, only the owner of a EHR can directly access the personal data. Since sensitive information and non-sensitive data are separated before data sharing, there is no chance of medical data leaking in PoF. Off-the-chain data storing is an iconic feature of this study. This architecture stores hashes of the locally stored personal information in the blockchain. Those hashes of locally stored personal information are shared with all decision-making entities. However, illegal modification of personal information attack can happen in two situations:

- The attack on blockchain data (on-the-chain data): If a node (attacker) attempts to modify a block illicitly, the consensus from all legitimate nodes can demolish the attack. Similarly, decentralized architecture helps to retrieve lost information or corrupted information from leftover nodes. Block data can also cross check with locally stored hash information of similar data [45].
- The attack on local data store (off-the-chain data): If a node (attacker) attempts to modify locally stored information and successfully changes (corrupts) the local database information, the node will not be corrupted and unknown for long. During collaborative medical decision-making, hashes of every local databases and blockchain databases are cross-checked before a decision is finalized. While initiating the next collaboration, hashes of the corrupted node and the rest of the nodes will be mismatched. Because of this off-the-chain data storing architecture, hashes of local and blockchain databases can be checked.

Therefore, off-the-chain blockchain successfully secures the integrity of the data to uphold the overall privacy of the proposed PoF based collaborative medical decision-making system [78].

### 4.2.9. Scalability

Scalability of the system includes adaptation capability of the number of users and nodes in a system. System load of the proposed PoF based collaboration is already minimized by storing EMR and EHR off-the-chain. In addition, use of consortium blockchain rises the system scalability.

### 4.2.10. Throughput Performance

As this is an ongoing work that is developing its own blockchain implementation platform, this study provides only an overview of system performance. In bitcoin and Ethereum, the block size is limited, and block time is higher. For example, in bitcoin block size is 1 MB and the average block time is 10 min. That is, including network constraints and limitations, average throughput of bitcoin is between 3.3 to 7 or $1,000,000/10/60/230 = 7.25$ tx/second (where a small transaction is assumed to 230 bytes) [41,42]. On the contrary, multichain does not restrict block size and time. Therefore, we can even set block size up 1 GB, and block time to 2 s. That is, including network constraints and limitations, average throughput of used multichain blockchain can be up to 2 million tx/second [79]. Therefore, block size and transmission latency are the major factors to decide overall throughput. Firstly, we are updating our decision judging factors to finalize our block size. Secondly, we are planning to test the PoF based scheme in a large environment (in the Inje University Hospital) [80]. So this study plans to include exact network performances (block size, block time) in the future. In addition, it plans to compare network performance of other studies as well.

### 4.2.11. Comparison with a Few Other Consensus Algorithms

Several consensus gathering algorithms are studied during PoF designing. Now, a theoretical comparison of PoF with existing consensus algorithms is presented (Table 5) based on the performance metrics of Hyperledger [81]. How the proposed PoF based collaborative medical decision gathering system is qualitatively superior to existing consensus gathering algorithms is elaborated here (based on the factors mentioned in Table 5):

- Node management: In between the consortium and permissionless blockchain, consortium blockchain is more controlled by the peers and allows comparatively easy modification of block information. If blockchain nodes are limited, identities of blockchain nodes can also be anonymous. Therefore, the proposed method chooses the consortium system for efficient node management. However, PBFT [37] and Tendermint [40] assess identities of the available nodes during a consensus mechanism. Alternatively, most of the permissionless consensus (PoW [35], PoS [38], and Ripple [39]), nodes are able to join the network without any identity verification [82].

- Transmission rate: The proposed system experiences a higher transmission rate for three reasons. Firstly, this scheme is a consortium blockchain with a limited number of nodes, so consensus gathering needs reasonably less time. Secondly, the use of off-the-chain data storing architecture also reduces the overall block size. Finally, Multichain [79] allows for the configuration of block size and block time, so the overall throughput increases in the proposed scheme by many times. Alternatively, PoS encourages the owner of the coin to prolong the coin holding time [83], so it also experiences a slightly higher transmission time. Similarly, as PBFT needs several transactions among peers before approving a block, transaction time is also high like Ripple [84]. As PoW demands higher node performance, the overall bottleneck increases [82]. Similarly, Ripple needs to wait till 80% are in agreement, and it also needs a higher transaction time [82]. On the other hand, the proposed PoF based collaborative demands 75% agreement (which is also changeable).

- Energy consumption: In PoW, mining reward depends on the amount of work done, so the computer system of the miners consumes higher energy. Compared to PoW, PoS consensus gathering algorithm requires less energy but is still high [82]. Alternatively, the proposed system (PoF) consumes less energy, as no reward system is available on work.

- Scalability: As PoW and PoS have higher energy consumption rate and block size, they preserve a higher scalability issue. PoS improved scalability by achieving better latency with a smaller block size [84]. Alternatively, due to off-the-chain data storing facilities and lower block size, the proposed PoF based collaborative system is highly scalable [85]. PBFT, Ripple, and Tendermint also face lower scalability to the existing system for network bottleneck [85].

- Finality process: Unlike the PoW, PoS, Ripple, and Tendermint blockchain consensus algorithm, PoF follows an immediate or absolute finality process. That is, just after the agreement from decision giving entities a new block with the collaborative medical decision is created. Alternatively, most consensus algorithms (PoW, PoS, Ripple, and Tendermint) follow a probabilistic finality process that waits until a certain period before a block gets finalized [82]. However, real-world business also demands immediate finality, as no one wants to take the risk of losing assets during the waiting time [82,84].

- Adversary tolerated power: Adversary tolerated power states the amount of control an attacker must have to regulate a blockchain network. A node in PoW needs at least 25% computer power to manipulate or create a block. Similarly, an attacker needs to control over 51% (more than half) of the stakes to control over the blockchain [84]. In comparison to those, the proposed PoF restricts adversary tolerated power to 75%. That is, at least 75% of the participated entities must agree to take control of a chain [72].

**Table 5.** Comparison of proof of familiarity (PoF) with other consensus algorithms.

| Algorithm / Fact | PoF (Proposed) | PoW [35] | PoS [38] | PBFT [37] | Ripple [39] | Tendermint [40] |
|---|---|---|---|---|---|---|
| Node management | Permissioned | Open | Both | Permissioned | Open | Permissioned |
| Transmission rate | Low | Low | Medium | High | High | Medium |
| Energy consumption | Low | High | Medium | Low | Low | Low |
| Scalability | High | Low | Low | Medium | Medium | Low |
| Finality process | Immediate | Probabilistic | Probabilistic | Immediate | Probabilistic | Probabilistic |
| Adversary tolerated Power | 0.75% diversity rate | < = 25% Computing Power | < 51% stake | < 33% Voting power | < 20% faulty nodes in UNL | < 33.3% Voting power |

*4.3. Limitation and Future Work*

Although the study covers most of the collaborative medical decision-making situations and factors, there are still a few more factors that will be considered in the future version of this study. A few limitations, drawbacks, and future plans of this study are discussed here.

4.3.1. Prioritization of Decision Giving Entities

This study introduced four decision-making entities (P, D, $I_c$, and $C_p$) for a collaborative medical decision gathering. Although, PoF evaluated all success factors of an entity, prioritization was not considered among the entities (P, D, $I_c$, and $C_p$). In this version of the study, the PoF consensus algorithm weighs each decision giving entities (P, D, $I_c$, and $C_p$) equally. That is, PoF emphasizes the decision of P, D, $I_c$, and $C_p$ equally during decision-making for the patient, P from P, D, $I_c$, and $C_p$. Although, in real life, the impact of an individual's decision might differ based on the entities' nature. However, to do this prioritization, concrete evidence and empirical analysis is needed. Therefore, the study acknowledges the issue and plans to consider this in the future version.

4.3.2. Identical IFI of the Decision Giving Entities

Equal value of prioritization evaluation factors within an entity might also give rise to conflict in PoF. Individual familiarity index (IFI) of the decision giving entities depends on decision judging factors. In some cases, IFI value can be identical for two of the same kind of entities. However, in the future PoF plans to handle the situation by prioritizing decision judging factors of each entity (P, D, $I_c$, and $C_p$). For example, if two doctors' (A and B) IFI value are x and x' (where x = x'), current PoF cannot handle that where job experience time ($J_{ET}$) and treatment success rate ($T_{SR}$) are decision judging factors. Current PoF evaluates both $J_{ET}$ and $T_{SR}$ equally, but in the future, PoF plans to weigh between $J_{ET}$ and $T_{SR}$. Therefore, equal IFI value (x = x') conflict can be solved after fine-tuning the mentioned issue.

4.3.3. The Inclusion of Critical Aspects of Medical Decision-Making

Collaborative medical decision-making is a hard nut to crack [1]. Economical and psychological effects can never be ignored. Similarly, the participant's must be well aware of other entities' knowledge and capacity. Emergency and context-aware decision assessment are not considered at this moment. In the future, the study plans to include the aforementioned minor factors in the PoF consensus gathering algorithm. As the experimental evaluation and comparison with other studies are ongoing, minor factors were overlooked in this version of the study.

4.3.4. Real-Life System Implementation

Blockchain technology faces scalability issues during real-life deployment [52]. Proposed PoF based blockchain technology is designed with the best possible scalability. As medical collaboration is not meant for a large group, proposed consortium blockchain can accommodate adequate participating entities. Another issue of blockchain is a large block size (storage issue) and transaction delay. The off-the-chain data storing used in this study reduced block size and transaction delay. This rapid transaction among peers increases the chance of commercial deployment of PoF based on collaborative medical decision-making. Last but not least, no reward is introduced in the PoF consensus algorithm. As IFI totally depends on the individual's achievement, no unwanted money and energy are spent on mining and computing. Therefore, PoF based collaborative medical decision-making system is scalable enough for commercial deployment. Alternatively, management of a local database for each participating entity (P, D, $I_c$, and $C_p$) can be difficult. In a future version, we plan to deliver in detail node management strategies designed for PoF based medical collaborative decision-making system.

4.3.5. Empirical Validation Comparison with Studies

One of the major limitations of this study was that it only provided a qualitative comparison among the proposed study and existing studies. This version of the study only tested with a prototype of the PoF based collaborative medical decision-making system with the help of third party (multichain) platform [72]. This work is an ongoing work, that includes the development of an independent blockchain framework followed by its performance analysis. Upon completing the whole system, this study plans to include real-time network behavior analysis of a PoF based collaboration system and existing medical decision gathering platforms. Because of this limitation, this study only includes theoretical comparison with existing platforms. At the same time, the study found strong qualitative advantages that encourages us to build the whole system in future. In the future version, the study plans to include critical blockchain system analyzing factors (network delay, block creation time, block finality time, energy consumption, etc.).

**5. Conclusions**

In this paper, we have proposed an off-the-chain consortium blockchain scheme for collaborative medical decision-making that includes the security of blockchain and privacy of personal data. We also have presented the proof of familiarity (PoF), a consensus gathering algorithm, designed to assimilate medical decisions of healthcare stakeholders (patient, doctor, insurance company, cured patient). Working mechanism and efficiency of the proposed PoF consensus algorithm is confirmed with an open source blockchain simulation platform (multichain 2.0). PoF ensures the integrity of a collective medical decision and privacy of stakeholders by storing previously stored decisions and identities by blockchain. While preserving the identity of stakeholders, PoF follows a two-layer security measure. Firstly, the identities of patients, doctors, cured patients, and insurance companies are locally stored, and secondly, the hash of those are stored in a block. Moreover, the qualitative performance of PoF is compared with the existing collaborative medical decision-making system, and theoretical comparison identifies PoF as an efficient use of blockchain in healthcare. PoF successfully reduced the risk of personal information (PII and PPII) leaking during collaboration of medical decisions. The following are the key outcomes we gathered from scheming and examining the PoF based collaborative medical decision-making system:

- PoF based collaborative medical decision-making system provides middlemen with a less decentralized medical decision collaboration platform with no single point of failure.
- PoF based collaborative medical decision-making system uses modified blockchain architecture (off-the-chain) to secure clinical data privacy and security.
- PoF based collaborative medical decision-making system allows trusted participation of significant medical decision-giving entities to provide a better clinical decision.

This study identified the reasons for the lack of medical collaboration and associated risks. Additionally, a PoF based collaborative scheme successfully mitigates ongoing issues of collaborative medical decision-making by using blockchain technology. Our future research directions include developing a fully-fledged real-time application, efficient use of previous decisions, and learning from the environment. In the future version of this study, we will consider the weighting factors among entities during collaborative decision-making. This shall then enable the system to be commercially used in a broader scope with more accuracy.

**Author Contributions:** Conceptualization, J.Y. and C.-S.K.; formal analysis, M.M.H.O.; methodology, N.-Y.L. and M.A.; project administration, J.Y. and C.S.-K.; supervision, N.Y.-L. and C.-S.K.; visualization, M.M.H.O.; writing—original draft, M.M.H.O.; writing—review and editing, M.A.

**Funding:** This research was funded by Institute for Information and Communications Technology Promotion (IITP) grant funded by the Korean government (Ministry of Science and ICT) (No. 2018-0-00261, GDPR Compliant Personally Identifiable Information Management Technology for IoT Environment). The APC was funded by Institute for Information and Communications Technology Promotion (IITP) grant funded by the Korean

government (Ministry of Science and ICT) (No. 2018-0-00261, GDPR Compliant Personally Identifiable Information Management Technology for IoT Environment).

**Acknowledgments:** The authors acknowledge computer and communication (CnC) lab members of Inje University for providing administrative and technical support. Finally, the authors thank anonymous reviewers for their valuable review.

**Conflicts of Interest:** The authors declare no conflicts of interest.

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
