# Peer review of "Proof-of-Familiarity: A Privacy-Preserved Blockchain Scheme for Collaborative Medical Decision-Making"

_applsci, doi:10.3390/app9071370_

Round 1

Reviewer 1 Report

Authors proposed a very interesting framework for better medical decision making. Before making final publication decision, the following issues are needed to be addressed:

Throughout the paper, many places, authors end sentences with "too", which needs to be addressed.

I do not think medical decisioning is an appropriate term. I suggest to replace it with "medical decision making"  in the entire paper.

Literature review section should be re-organized for better readability with natural flow, identify the gap, and better explain how this will be filled by this study.

Consensus gathering algorithm needs better explanation about how to handle conflict (with what objective) in the event of multiple experts (doctors)/patients engagement. Barriers/feasibility to implement the proposed algorithm is also required to address.

Author Response

First of all, the author would like to express deep gratitude for critically reviewing the manuscript. We concur with your valuable assessment and carefully considered your comments to revise our manuscript. Revised meant for you are highlighted with pink colour throughout the manuscript. Based on your review, the following changes have been made:
1. The revised version of the manuscript rephrases all the sentences those end with “too”.
2. The term ‘medical decisioning’ was substituted with the term ‘medical decision-making’.
3. The authors totally agree with the reviewer’s opinion about the “literature review” section. Therefore, the updated version of the manuscript removed redundant parts. In addition, the literature review section has been reordered for better readability and harmonization. This version of the manuscript critically analyzed several existing studies on blockchain based medical decision making. In addition, the current version of the manuscript identifies issues of those studies followed by the solutions proposed by the PoF based collaborative medical decision-making scheme. (Page Numbers: 4, 5 and 6)
4. Firstly, the authors thank the reviewer for recognizing this flaw of the proposed scheme. An efficient collaborative medical decision-making scheme requires 360-degree evaluation. Therefore, the PoF consensus algorithm considers all possible scenarios for developing a successful medical decisionmaking scheme. As the work is continuing, some minor factors might still be overlooked.
This version of the manuscript mentions includes special scenarios such as conflicting among decision giving entities in detail and confusion during equal individual familiarity index (IFI). (Page Number: 18)
As the work is still ongoing, the study is evaluating the weighting factors of considered decision judging factors. At the same time, this version includes a few conflicting scenarios and possible results of PoF consensus algorithm. (Page Numbers: 18)
Along with these factors, system implementation limitations and feasibly of the proposed scheme were critically analyzed in the revised version of the manuscript. This version also includes a new section ‘Limitation and Future work’ that explains detail drawback and future plan. (Page Numbers: 18 and 19)

We hope that the aforementioned replays satisfy your concerns and improve overall manuscript quality. Please let us know if you have any other recommendations and suggestions.

Reviewer 2 Report

The study proposes a collaborative medical decision gathering private-blockchain scheme and associated proof of familiarity (PoF) consensus algorithm.

The main drawback of the study is the lack of comparison of the proposed PoF against other consensus mechanisms. Table 5 compares these algorithms; however, authors need to describe more in detail each fact (Node management, transmission rate, etc) to prove that their solution provides, for example, lower energy consumption than PoS.

As authors have implemented their algorithm in Multichain 2.0, I encourage the authors to provide some quantitative analysis of the performance of the networks, in terms of delays, transmission rate, block generation rate, costs, etc.

Some question:

What is the meaning of off-chain blockchain (increase confidentiality, line 110)? I suppose the authors mean off-chain transactions. However, how off-chain transactions increase confidentiality. This must be explained in the paper.

A similarity check by using the Turnitin tool results in 10 % of content similar to other Internet sources (excluding references). This ensures the originality of the manuscript.

Just consider rephrasing lines from 174 to 182, as they are similar to a paper submitted to Korea Advanced Institute of Technology.

Some minor corrections:

middleman less transaction -> improve the definition of bitcoin

blockchain based -> blockchain-based

Line 201: Figure 6. -> Figure 6

Line 289: by algorithm 1 -> by Algorithm 1.

Line 343: an open source blockchain implementation platform -> an open source blockchain implementation platform,

Please follow MDPI template:

Sect. 2 -> Section 2

Author Response

We concur with your valuable assessment and carefully considered your comments to revise our manuscript. Please consider the revised texts that are highlighted with Turquoise colour in the updated manuscript. Based on your review, the following changes have been made:
1. The authors appreciate your constructive criticism. This version of the manuscript elaborates every considered factor of Table 5 (Comparison Table). The authors discussed the superiority of the proposed scheme in comparison to other consensus gathering schemes. This version includes in detail elaboration of the proposed scheme on ‘Node management, Transmission rate, Energy consumption, Scalability, Finality process, adversary tolerated power’ etc. The authors highlighted the key contribution of the proposed scheme to overcome issues prevailed in existing studies (Page Numbers: 16 and 17)
2. The authors concur with the reviewer comments that a limitation of this study is the study only provides a qualitative comparison with existing schemes. The authors identified the following reasons behind this:
a. This version of the study has tested a prototype of PoF based collaborative medical decision-making-system which is a third-party blockchain implementation platform (Multichain). Although initially third-party test environment was fine, but the authors currently working to develop an independent blockchain platform in order to test explicitly. Upon completing the whole system, this study plans to include real-time network behaviour analysis of PoF based collaboration system and existing medical decision gathering platforms as well. The current version includes basic throughput
characteristics of the proposed schemes. (Page Number: 16; Line 494 to 506).
In future, the study plans to test the PoF based scheme in a larger environment (preferably, Inje
University hospital).
b. As the study includes sensitive health information and decision, the authors must acquire an
institutional review board (IRB) permission to disclose the EMR and EHR. The authors are currently
working to receive that permission. In future, the authors plan to include full empirical evaluation
of the study.
c. Finally, the authors have shown adequate qualitative evidence to justify the proposed PoF based
collaborative decision. The authors added ‘Limitation and Future work’ section in this version to

discuss the aforementioned issues in detail. (Page Numbers: 18 and 19).
In addition, ‘Throughput performance:’ section is also added to discuss the overview of the
throughput performance of the proposed scheme. (Page Number: 16)
3. The authors include a separate section that discusses how ‘off-the-chain’ blockchain increases the confidentiality/privacy of our proposed scheme. In addition, this study explains two privacy leaking scenarios and associated protection served by ‘off-the-chain’ blockchain. However, off-the-chain blockchain includes both off-the-chain storing and off-the-chain transaction. The authors also include references to other studies where off-the-chain data storing and transaction was used to increase system privacy. (Page Numbers: 15 and 16)
4. The authors feel extremely sorry as the lines 174 to 182 are matched with another work of us (checked from KAIST), However, most parts of the literature review section were rearranged and rewrite in this version of the manuscript. This includes the rephrasing of lines from 174 to 182 as well. (Page Numbers: 5 and 6)
5. This version of the manuscript includes improves the definition of bitcoin with necessary references. (Page Number: 4)
6. Throughout this updated manuscript, the term ‘blockchain based’is changed to ‘’blockchain-based. Similarly, few other grammatical errors were also corrected all over the manuscript. For example, the term ‘decision making’ is changed into ‘decision-making’.
7. The current version uses ‘Figure 6’ instead of ‘Figure 6’.
8. Updated manuscript uses ‘Algorithm 1’ instead of ‘algorithm 1’.
9. This version of the manuscript uses ‘an open source blockchain implementation platform,’ ‘instead of an open source blockchain implementation platform’.
10. Thanks again for this comment. This version is updated according to MDPI template. For example, ‘Sect. 2’ was replaced by ‘Section 2’.

The authors tried to cover most of your concerns to update the manuscript. The authors, therefore, hope that the aforementioned replays satisfy your concerns and improve overall manuscript quality. Please let us know if you have any other recommendations and suggestions.

Round 2

Reviewer 1 Report

Authors addressed all the comments 

Reviewer 2 Report

The authors have provided a substantially revised version of their manuscript. My appreciations have been considered and the paper has been improved with these new changes.

I think that the paper is ready to be accepted.